

# The Zoige pioneer plant *Leymus secalinus* has different endophytic bacterial community structures to adapt to environmental conditions

Yue Xia, Ruipeng He, Wanru Xu and Jie Zhang

College of Life Sciences, Sichuan University, Chengdu, Sichuan, China

Corresponding author
Jie Zhang, zhangjfy@scu.edu.cn

## ABSTRACT

**Background.** *Leymus secalinus* is a pioneer plant grown in the Zoige desertified alpine grassland and it is also one of the dominant plant species used for environmental remediation. *L. secalinus* plays a large role in vegetation reconstruction in sandy land, but the abundance and diversity of its endophytes have not yet been investigated.

**Objectives.** This study was performed to investigate the changes in the endophytic bacterial community structure of *L. secalinus* under different ecological environments and to analyze the effects of environmental changes and different plant tissues on the *L. secalinus* endophytic bacteria.

**Methods.** Leaf, stem, and root tissue samples of *L. secalinus* were collected from Zoige Glassland (Alpine sandy land) and an open field nursery (Control). DNA was extracted and the 16S ribosomal DNA was amplified. The sequence library was sequenced on an Illumina MiSeq platform and clustered by operational taxonomic units (OTUs). $\alpha$-diversity and $\beta$-diversity analyses, species diversity analyses, functional prediction, and redundancy (RDA) analyses for the soil physicochemical properties were conducted.

**Results.** $\alpha$-diversity and $\beta$-diversity analyses showed that the endophytic bacteria in *L. secalinus* varied in different areas and tissues. The abundance of *Allorhizobium-Neorhizobium-Pararhizobium-Rhizobium*, which is related to nitrogen fixation, increased significantly in the *L. secalinus* found in the Zoige Grassland.

Moreover, the abundance of nutrition metabolism and anti-stress abilities increased in functional prediction in the desert samples. The soil physicochemical properties had an insignificant influence on bacterial diversity.

**Conclusion.** The changes in the endophytic bacterial community structure in *L. secalinus* were significant and were caused by environmental alterations and plant choice. The endophytic bacteria in *L. secalinus* grown in alpine sandy land may have greater anti-stress properties and the ability to fix nitrogen, which has potential value in environmental remediation and agricultural production.

## INTRODUCTION

The Zoige alpine grassland, located on the eastern edge of the Qinghai Tibet Plateau (QTP), is an important part of the Jeurre Prairie in China. Its altitude is approximately 3,500 m, and its monthly average temperature is −9.6 °C to 11.2 °C. It is a water conservation area of the Yellow River and the Yangtze River, and is important in maintaining the area's ecological balance and animal husbandry. Owing to climate changes (*e.g.*, increasing temperature and decreasing precipitation) (*Hu et al., 2015*; *Niu, Ma & Zeng, 2008*) and anthropogenic factors (*e.g.*, over-grazing, drainage of water systems) (*Dong et al., 2009*), degradation of these grasslands began in the 1970s and peaked in 2000 (*Hu et al., 2018*). The degradation of the grasslands was controlled by the establishment of biological and sand barriers, as well as a prohibition on cattle grazing (*Liu et al., 2020*; *Luo et al., 2019*; *Yan & Lu, 2015*). The Zoige Grassland has been increasingly used as an experimental site for the establishment of multiple ecosystem governance (*Jian et al., 2022*; *Liu et al., 2020*; *Luo et al., 2019*).

The Zoige Grassland is characterized by high-altitude, cold temperatures, and barren soil, all of which present problems for the reconstruction of vegetation. *Leymus secalinus* is a pioneer plant grown in the Zoige's sandy soil due to its powerful vitality. Moreover, *L. secalinus* has the ability to meet the nutritional requirements of livestock. *L. secalinus* is a perennial herb belonging to the family Poaceae, the dominant species that grows in active dunes (*Kang, Zhao & Zhao, 2017*). *L. secalinus* reproduce both sexually through seed and clonally *via* the vegetative growth of rhizomes that can develop into a new individual in a short time. These characteristics make it beneficial for the rapid remediation of sandy lands. The proportion of spreading ramets in *L. secalinus* is increased in low-nutrient soils (*Ye, Yu & Dong, 2006*), and the rhizome biomass, length, diameter, and adventitious roots at the rhizome nodes are enhanced under drought conditions (*Zheng et al., 2021*). Furthermore, all of the ramets are changed as described above owing to clonal integration (*Sui et al., 2011*), which helps them to manage stressful conditions. These qualities make *L. secalinus* an ideal plant for phytoremediation in alpine sandy land.

The endophytic bacteria found in plants may play an important role in stress resistance, during periods of salt stress (*Khan et al., 2020*), low temperature (*Subramanian et al., 2015*), and drought (*Ullah et al., 2019*). Moreover, many studies have shown that endophytic bacteria can promote plant growth (*Adhikari & Pandey, 2020*) and increase the absorption of mineral elements (*Afzal et al., 2019*) by the host to enhance its viability. Plants may develop unique endophytic communities in environmental extremes (*Przemieniecki et al., 2019*; *Yang et al., 2020*), which are highly valuable for research and application.

Many studies have shown the symbiotic relationship between bacteria and plant tissues. Our understanding of endophytic bacteria has improved due to high-throughput technology. With it we have also discovered a two-way relationship between endophytic bacteria and plants. For example, the plant roots secrete compounds that attract or repel bacteria, and the bacteria secrete pectinases or cellulases to colonize the roots and even move to the stem (*Santi, Bogusz & Franche, 2013*). The diversity of plant endophytic bacteria is related to the plant species (*Jian et al., 2022*), soil type (*Ling et al., 2020*), growth stage (*Mohr et al., 2021*), and climatic conditions (*Zhang et al., 2018*). The understanding of endophytic

bacteria is advancing rapidly with the development of high-throughput sequencing. Research on the endophytic bacteria of plants on the QTP's unique environments has received great attention. Previous studies have focused on the unique and dominant plant species of an area (*Liang et al., 2021*; *Wei et al., 2022*; *Yang et al., 2020*). However, the plants used in phytoremediation in the QTP have yet to be adequately studied.

Endophyte diversity and biological functions are often distinctive under high-altitude, cold, and barren conditions. For example, *Burkholderia* is highly abundant in the arctic pioneer plant mountain sorrel (*Oxyria digyna*). Many studies have proven that *Burkholderia* has a variety of growth-promoting functions, such as nitrogen fixation (*Bhattacharjee, Singh & Mukhopadhyay, 2008*) and phosphate solubilization (*Kour et al., 2021*). In another study, the seed bacterial community of eight native alpine grassland plants growing in the QTP was dominated by $\gamma$-Proteobacteria, which can comprise up to 99% of the endophytic community (*Liang et al., 2021*). Based on these characteristics, we explored the endophytic bacterial community structure of *L. secalinus* in different ecological environments. The effects of environmental changes and various plant tissues on the endophytic bacteria of *L. secalinus* was also studied. These variations may explain the vitality of *L. secalinus* grown in the Zoige alpine sandy soil and provide a resource for environmental remediation and agricultural production.

## MATERIALS & METHODS

### Collection and preparation of sample materials

The *L. secalinus* (Georgi) Tzvelev collected from the Zoige alpine sandy soil was designated DL, and the samples collected in the open field nursery in the laboratory were designated L. The sampling site in the Zoige sandy soil (Fig. 1) is located in the Zoige alpine grassland (33°41′0″N, 102°56′7″E; altitude: 3,500 m), which has a mean annual precipitation >600 mm. The average monthly temperature during the winter ranges from −9.6 to −3.9 °C. The open field nursery is located at Sichuan University (Chengdu, China) (30°38′2″N, 104°4′35″E; altitude: 500 m) with the mean annual precipitation >800 mm; the average monthly temperature during the winter ranges from 5.6 to 7.9 °C. The entire plant was dug up at each location, and the rhizospheric soils were removed. Five plants were randomly selected and considered to be one sample. Three parallel samples were collected from the roots (LR and DLR, without rhizomes), stalk (LS and DLS), and leaves (LL and DLL) and then stored in liquid nitrogen.

The plant tissues were surface sterilized in a laminar flow hood. The plant tissues were sterilized for 30 s in 75% ethanol, washed with sterile water once, and then soaked in 0.1% mercuric chloride ($HgCl_2$) for 5 min. After that, the plant tissues were washed through surface sterilization three times with sterile water. Water from the final wash was plated on NB agar plates to ensure that the surface was sterilized.

### DNA extraction

All the DNA of samples was extracted using a PowerSoil DNA Isolation Kit (MoBio Laboratories, Carlsbad, CA, USA) following the manufacturer's instructions. Then, the above genomic DNA was developed on 1% agarose gels using a NanoDrop

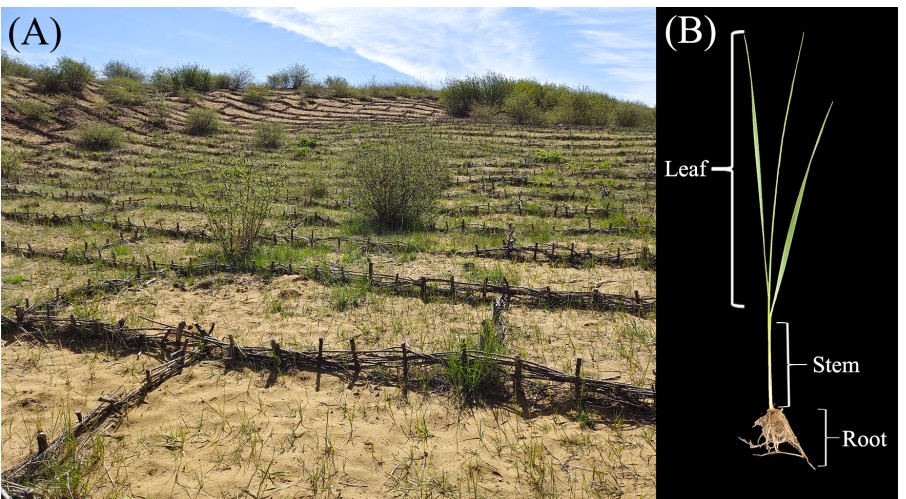

**Figure 1** Sampling location and organism.

**Table 1** All the primers used in this study.

| Organ | Position | Direction | Primer name | Sequence 5′–3′ | Reference |
|---|---|---|---|---|---|
| Root | 16S V5-V7 | Forward | 779F | AACMGGATTAGATACCCKG | *Chelius & Triplett (2001)* |
| | | Reverse | 1193R | ACGTCATCCCCACCTTCC | *Bodenhausen, Horton & Bergelson (2013)* |
| Stem and leaf | 16S V3-V4 | Forward | 332F | ACGGHCCARACTCCTACGGHA | *Chen et al. (2022)* |
| | | Reverse | 796R | CTACCMGGGTATCTAATCCKG | |

spectrophotometer (Thermo Fisher Scientific, Waltham, MA, USA) to ensure a high purity and quality.

## PCR amplification

All of the primers used in this study are shown in Table 1. The 5′ end of the forward and reverse primers had an 8-digit barcode sequence added to differentiate the different samples. The PCR was performed as previously described using 25 µl reaction volumes on a Mastercycler Gradient (Eppendorf, Hamburg, Germany). The reaction volumes followed those used by *Lu et al. (2022)*. The cycling parameters were modified slightly from *Chen et al. (2022)*, which included 95 °C for 5 min to lead to the DNA melting, followed by 30 cycles of 95 °C for 20 s, 55 °C for 30 s, and 72 °C for 30 s, with a final extension at 72 °C for 5 min.

## High-throughput sequencing and data analyses

Deep sequencing was performed on the Illumina MiSeq platform (Illumina, San Diego, CA, USA) at the Allwegene Company (Beijing, China). The above amplicons were pooled to an equimolar ratio. According to standard protocols, paired-end sequencing was performed on the Illumina MiSeq (2 × 300). Raw paired-end reads were joined to generate clean merged reads after filtering adaptors, low-quality reads, host sequences, and barcodes.

All of the analysis are based on clean merged reads. Clean merged reads were clustered into OTUs using the UPARSE algorithm of Vsearch (v2.7.1) with a 97% similarity (*Edgar, 2013*). All of the OTUs were sorted into different taxonomic groups using the Ribosomal Database Project (RDP) classifier tool against the SILVA128 database (*Cole et al., 2009*).

The species abundance bar plot diagram analysis was performed using R (v3.6.0) and based on taxonomic annotation and relative abundance results. The principal coordinates analysis (PCoA) and principal coordinate analysis (PCA) were also analyzed using R (v3.6.0) and were based on the OTUs collected from each sample to examine the similarity between different samples (*Wang et al., 2012*). The unweighted pair group method with arithmetic mean (UPGMA) clustering tree described the dissimilarity (1-similarity) between multiple samples. These values, based on the evolutionary distances between the microbial communities from each sample, were calculated using Bray-Curtis algorithms (*Jiang et al., 2013*).

### LEfSe analysis

Linear discriminant analysis (LDA) effect size (LEfSe) (*Segata et al., 2011*) (LDA score >4) was performed with the LEfSe tool to identify species that have significantly different abundances between groups.

### Function prediction

16S rRNA sequences from the Kyoto Encyclopedia of Genes and Genomes (KEGG) database was used for functional prediction in PICRUSt2. Using the information in the KEGG database, the abundance of each functional category for all samples was illustrated as heatmaps (*Hidalgo et al., 2021*).

### Soil physicochemical properties

The physicochemical properties of the non-rhizosphere soil from the Zoige Grassland and control samples from the laboratory were documented to evaluate the influence of soil nutrient conditions on the endophytic bacteria. The soil pH was determined with a ratio of 1:5 soil: ultrapure water (v/v) using a PHS-3S pH-meter (INESA & Scientific Instrument, Shanghai, China) (*Jian et al., 2022*). The soil organic carbon (SOC), total nitrogen (TN), total phosphorus (TP), total potassium (TK), available nitrogen (N), available phosphorus (AP), and available potassium (AK) were determined following *Zhang et al. (2015)*. Indophenol blue colorimetry was used to determine the ammonium nitrogen ($NH_4$-N) content, and dual-wavelength ultraviolet spectrophotometry was used to determine nitrate nitrogen ($NO_3$-N). The relationship between the endophytic bacterial community abundances and the soil physicochemical properties was measured using redundancy analysis (RDA).

## RESULTS

### Analysis of the sequencing data

A total of 2,219,372 raw reads were obtained from 18 samples of *L. secalinus* (roots, stems, and leaves). After adjusting for quality control filtration and removing chimera,

**Table 2** $\alpha$-diversity of endophytic bacteria in DL and L.

| Group name | High-quality sequence | OTUs | Chao1 index | Coverage | Shannon index | Simpson index |
|---|---|---|---|---|---|---|
| LR | 162795 ± 100359[b] | 1500 ± 52.4[d] | 2740 ± 204.3[d] | 0.89 ± 0.0100[a] | 9.00 ± 0.084[d] | 0.99 ± 0.0000[a] |
| DLR | 259121 ± 67815[c] | 1209 ± 143.5[c] | 2208 ± 234.9[c] | 0.91 ± 0.0115[b] | 8.29 ± 0.450[cd] | 0.99 ± 0.0058[a] |
| LL | 41912 ± 4054[a] | 369 ± 23.3[b] | 476 ± 18.8[b] | 0.99 ± 0.0000[c] | 6.89 ± 0.121[bc] | 0.97 ± 0.0058[a] |
| DLL | 38072 ± 12171[a] | 349 ± 42.5[b] | 448 ± 54.3[ab] | 0.99 ± 0.0058[c] | 6.11 ± 1.755[ab] | 0.89 ± 0.1704[a] |
| LS | 43034 ± 13666[a] | 286 ± 23.1[b] | 347 ± 25.8[ab] | 0.99 ± 0.0000[c] | 5.63 ± 0.596[ab] | 0.95 ± 0.0200[a] |
| DLS | 72405 ± 22092[ab] | 149 ± 11.5[a] | 218 ± 34.3[a] | 0.99 ± 0.0058[c] | 4.55 ± 1.006[a] | 0.86 ± 0.1617[a] |

**Notes.**
Data are presented as mean ± SE, ($n = 3$). Small letters reflect significant differences between different plant tissue samples ($p < 0.05$).
OTUs, operational taxonomic units.

non-specific amplicons, and host genes, a total of 1,852,014 high-quality sequences were produced. Each sample had an average of 102,889 reads. A total of 11,736 OTUs were obtained from high-quality sequences with a 97% similarity, and the number of OTUs of each sample ranged from 140 to 1,591 (Table 2).

### $\alpha$-diversity analysis

The $\alpha$-diversity of all the samples is shown in Table 2. All of the samples of plant tissue had a mean coverage index above 85%, which suggests that the samples were sequenced to an adequate depth. The Chao1, Shannon, and Simpson indices showed significant differences in *L. secalinus* endophytic bacteria in different ecological environments. The $\alpha$-diversity of the L group was higher compared with that of the DL group in the roots, stems, and leaves; this may be due to a higher level of nutrients available in L group soil (*Shi et al., 2014*). An increase in $\alpha$-diversity indicates high stability, which is more favorable for plant growth. In different tissues, the Chao1 and Shannon indices of the root samples (LR and DLR) were significantly higher than those of the stems (LS and DLS) and leaves (LL and DLL). This suggests that the endophytic bacteria in the roots had a greater diversity and mean abundance compared with those in the stems and leaves. Most endophytic bacteria can enter plants through the roots and then move toward the stems and leaves (*Santi, Bogusz & Franche, 2013*).

### $\beta$-diversity analysis

The PCoA analysis is shown in Fig. 2A. These results indicate that the distance between the roots and stems and leaves was much longer than the distance between the stems and leaves. We hypothesized that most of the endophytic bacteria of the stems and leaves spread from the roots. There is an enormous difference between the roots and stems, and leaves, as evidenced by the endophytic bacteria that moved and may be screened by *L. secalinus*. The hierarchical clustering results (Fig. 2C) agreed with those of the PCoA, which showed DLR and LR in the same branch while the others were in another branch.

Figure 2B describes the PCA analysis based on the abundance of OTUs, which noted that the L and DL groups could be differentiated from the coordinate axes. All of the DL groups were on the positive PC2 axis, and the L groups were on the negative PC2 axis. Although the PCA analysis is based on Euclidean distances, it is consistent with the

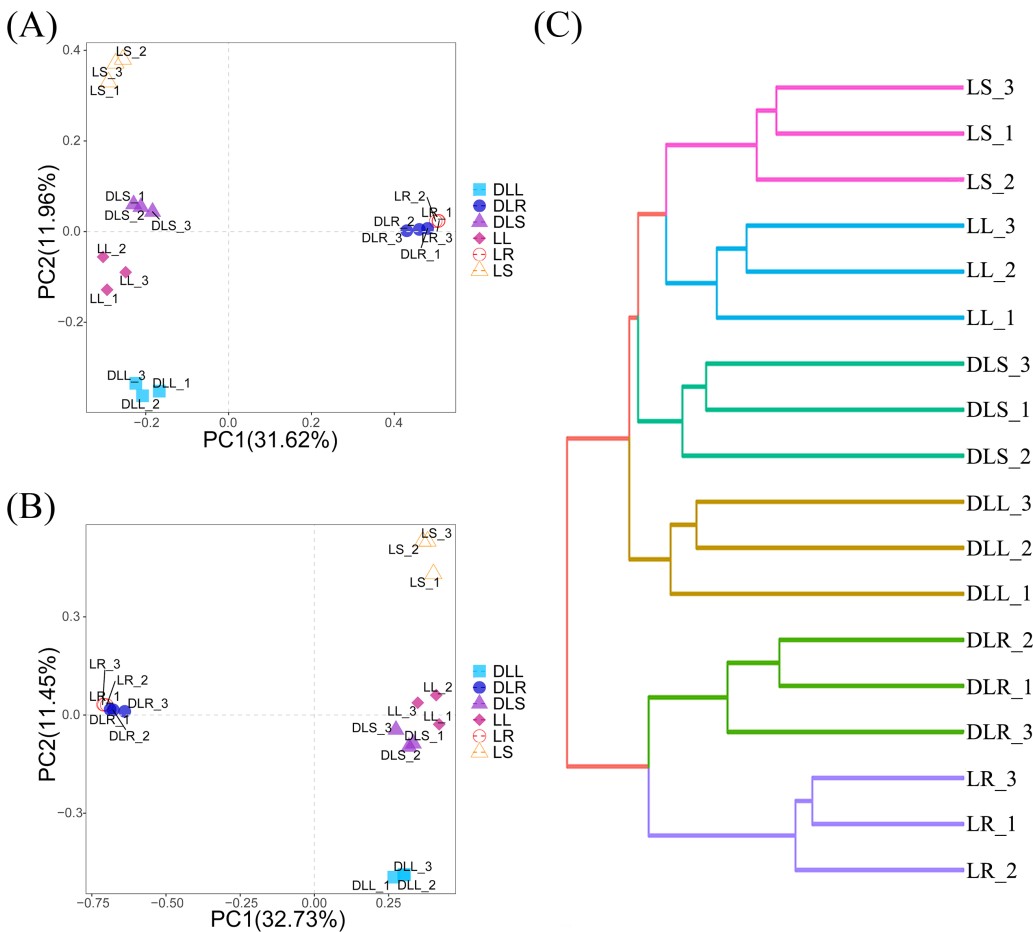

**Figure 2** *β*-diversity data. (A) Scatter plot of PCoA based on UniFrac and Bray-Curtis. (B) Scatter plot of the PCA score based on the Euclidean distance. (C) Hierarchical clustering. PCA, principal component analysis; PCoA, principal coordinates analysis.

results of PCoA, which also suggested that the endophytic bacteria in the roots differed substantially compared with those from the stems and leaves. This significant difference could have resulted from the different primers used and the selectivity of *L. secalinus*.

## Species diversity
### Bacterial community structure and diversity
Based on the OTU data, species abundance bar graphs (Fig. 3) were plotted using the R language tool. The endophytic bacteria for the DL group were classified into 35 phyla, 81 classes, 189 orders, 264 families, and 465 genera. The L group was classified into 35 phyla, 82 classes, 192 orders, 270 families, and 500 genera.

At the phylum level (Fig. 3A, Table S1), the endophytic bacteria were primarily Proteobacteria. The proportions of Proteobacteria in the DLR, DLS, and DLL samples were 44.0%, 46.3%, and 39.6%, respectively. The proportions in LR, LS, and LL samples were 48.3%, 84.3%, and 58.2%, respectively, which were similar to those of rice (*Oryza sativa*) as shown by *Kumar et al. (2021)*. The other major phyla of all the samples
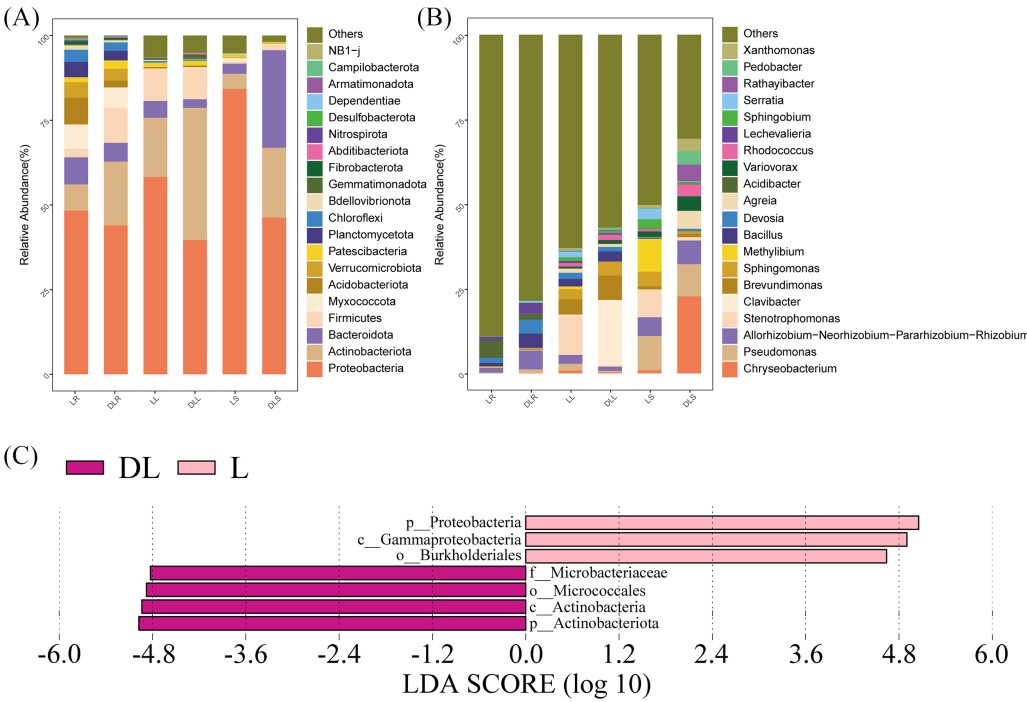

**Figure 3** **Relative abundance of endophytic bacteria.** Endophytic bacteria communities associated with *Leymus secalinus* at the phylum level (A) and genus level (B) and LEfSe analysis between DL and L groups (C). LEfSe, linear discriminant analysis (LDA) effect size.

included Actinobacteriota, Bacteroidetes, Firmicutes, Patescibacteria, Myxococcota, and Gemmatimonadota. The abundance of Actinobacteriota in the DLR and DLS increased significantly compared with those of the LR and LS.

At the genus level (Fig. 3B, Table S2), *Pseudomonas*, *Stenotrophomonas*, *Methylibium*, *Allorhizobium-Neorhizobium-Pararhizobium-Rhizobium*, *Sphingomonas*, *Brevundimonas*, *Acidibacter*, *Serratia,* and *Sphingobium* had relative abundances of more than 1.5% in the L group. Moreover, *Chryseobacterium*, *Pseudomonas*, *Allorhizobium-Neorhizobium-Pararhizobium-Rhizobium*, *Clavibacter*, *Brevundimonas*, *Bacillus*, *Devosia*, *Agreia*, *Variovorax*, *Rhodococcus*, *Lechevalieria*, and *Sphingomonas* had relative abundances of more than 1.5% in the DL group.

### Changes in the bacterial community structure

The amounts of Proteobacteria and Actinobacteriota were significantly different between the DL and L groups (Fig. 3C). We found that the relative abundance of Proteobacteria was reduced in all samples of the DL group compared with the L group (Fig. 3A, Table S1), and its abundance was significant in the stems ($P = 0.036$). The relative abundance of Actinobacteriota increased in all samples of the DL group compared with the L group (Figs. 3A and 3C, and Table S1), and it was significant in the roots ($P = 0.006$) and stems ($P = 0.047$). For the other phyla, Firmicutes significantly increased ($P = 0.024$), and

**Table 3   Physical and chemical properties of the soil.**

|  | pH | SOC (g/kg) | TN (g/kg) | TP (g/kg) | TK (g/kg) | N (mg/kg) | AP (mg/kg) | AK (mg/kg) | $NH_4^+$ (mg/g) | $NO_3^-$ (mg/kg) |
|---|---|---|---|---|---|---|---|---|---|---|
| L | 8.1 | 52.3 | 4.15 | 1.13 | 3.62 | 174.9 | 63.10 | 117.3 | 20.3 | 27.06 |
| DL | 8.5 | 2.61 | 0.25 | 0.80 | 2.14 | 7.67 | 20.82 | 39.14 | 0.69 | 7.71 |

Notes.

AK, available potassium; AP, available phosphorus; N, available nitrogen; $NH_4^+$, ammonium; $NO_3^-$, nitrate; SOC, soil organic carbon; TK, total potassium; TN, total nitrogen; TP, total phosphorus.

Acidobacteriota significantly decreased ($P = 0.039$) in the DLR compared with the LR (Fig. 3A, Table S1).

A series of changes at the phylum level could be related to the nitrogen balance. A study by *Fonseca et al. (2018)* showed that *Eucalyptus*, which was reported to have a negative N balance, had more than 50% Actinobacteriota in its rhizosphere microbiome. They also found that Firmicutes decreased and Proteobacteria increased as the soil's N content increased through the application of nitrogen or culture with legumes. The relationship between the N content (Table 3) and bacterial abundance in this study agreed with their results. Acidobacteria was reported to have a poor ability to fix nitrogen or solubilize phosphate, which may be due to alterations in the ecological environment.

At the genus level, the abundances of *Clavibacter* and *Brevundimonas* increased in the DLL compared with the LL, while the abundances of *Agreia*, *Chryseobacterium*, *Variovorax*, and *Rhodococcus* increased in the DLS compared with the LS (Fig. 3B, Fig. S1, and Table S2). The abundances of *Allorhizobium-Neorhizobium-Pararhizobium-Rhizobium*, *Polaromonas*, *Nocardioides*, *Microbacterium*, and *Paenibacillus* increased in the DLR compared with the LR (Fig. 3B, Fig. S1, and Table S2). All the genera described above were common among endophytic bacteria, and most of them contained species that could fix nitrogen or solubilize phosphate.

## Function prediction

The functions were predicted based on the KEGG database that utilized the 16S rRNA sequencing data (Fig. 4). We found that the DL group had a higher abundance of starch and sucrose metabolism than the L group, which was due to its low content of organic carbon (Table 3). In addition, the abundance of peroxisome and nucleotide excision repair in the DL group was greater than that in the L group, which was related to the ecological environment. The DL group was collected from the Zoige alpine sandy soil, which was exposed to strong UV rays and low temperatures. The endophytic bacteria in that environment had to respond to cell damage caused by stress. Regarding cell movement, the bacteria in the DL group moved substantially less than those in the L group, as reflected by its flagellar assembly and bacterial chemotaxis. In previous studies, bacteria have had increased locomotor activity under low nutrient conditions in order to obtain more nutrients (*Amsler, Cho & Matsumura, 1993*). However, we found that the functional abundance cell movement was reduced under low nutrient conditions (DL group) (Fig. 4 and Fig. S2), which may be due to insufficient nutrition to fuel movement in those bacteria.

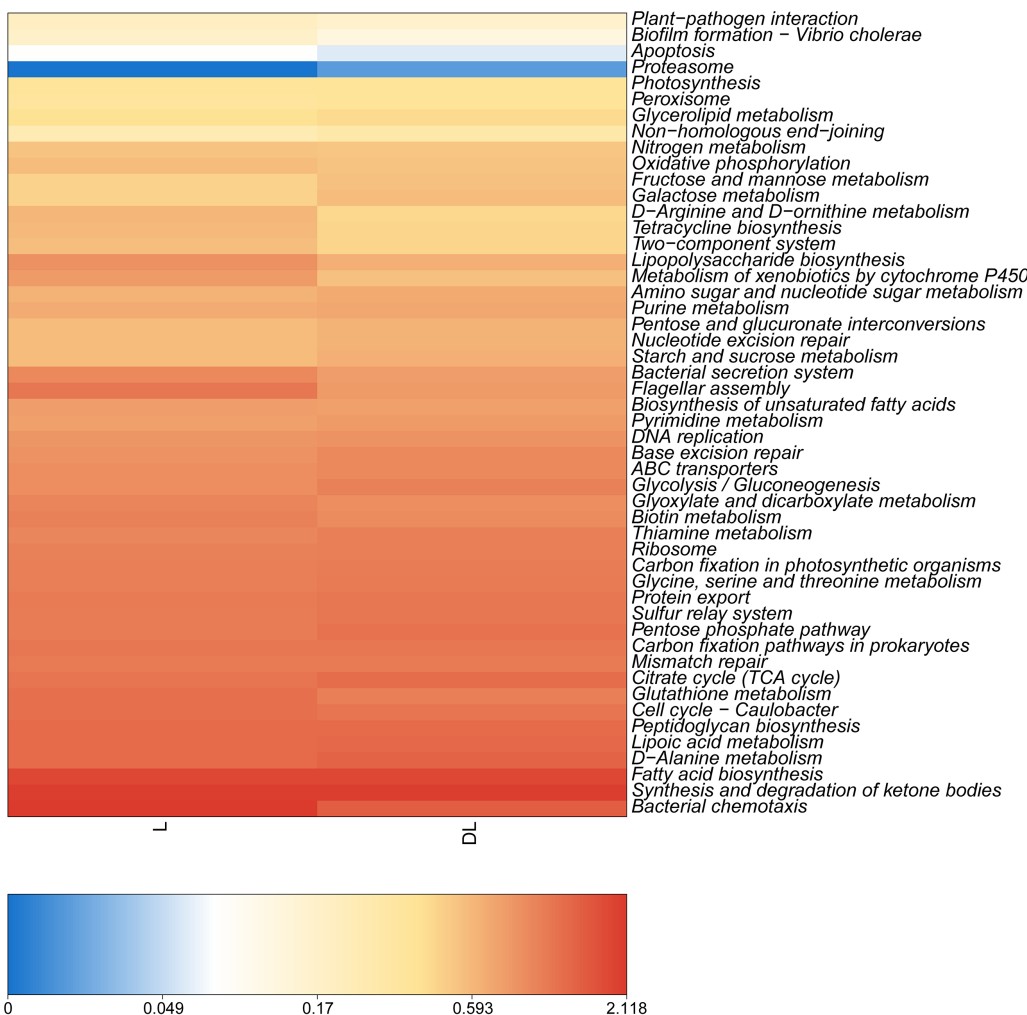

**Figure 4** Abundance heatmap of the KEGG function of endophytic communities. KEGG, Kyoto Encyclopedia of Genes and Genomes.

## RDA analysis of soil physicochemical properties

The relationship between soil physicochemical properties and species abundance was analyzed using an RDA analysis. The soil physicochemical properties whose *p*-values were less than 0.01 were filtered out, as shown in Fig. 5. At the phylum level (Fig. 5A), RDA1 and RDA2 only explained 5.86% of the variation in the endophytic bacterial community structure, and RDA1 and RDA2 explained 8.92% of the variation in the endophytic bacterial community structure at the genus level. We considered that the soil physicochemical properties did not adequately interpret the changes in the bacterial community. In our previous study, the soil physicochemical factors explained 93.5% (phylum level) and 93.47% (genus level) of the variation in the rhizosphere bacterial community structure (*Jian et al., 2022*). The RDA results may indirectly prove that the effects of the selectivity

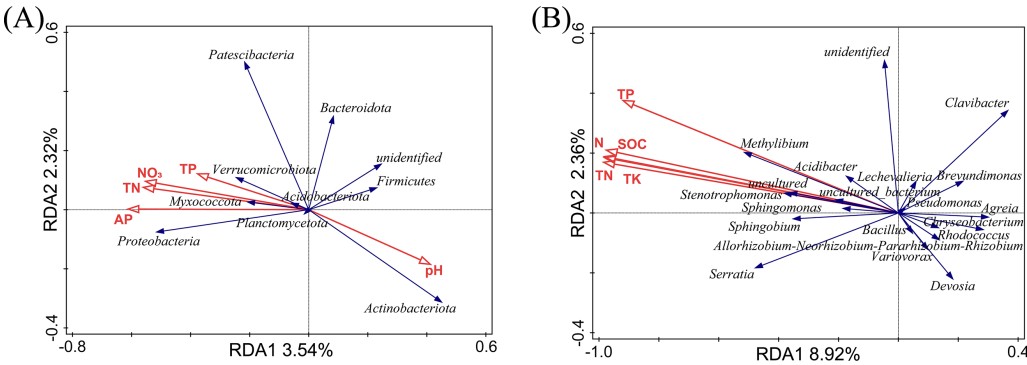

**Figure 5   RDA analysis at phylum level (A) and genus level (B).** RDA, redundancy analysis.

of plants were more potent than those of the soil physicochemical factors on the bacterial community in *L. secalinus*.

At the phylum level (Fig. 5A), the contribution of pH, TP, AP, NO$_3^-$, and TN were 31.2%, 26.7%, 26.0%, 14.0%, and 2.0%, respectively. The proteobacteria were the most affected by AP. Actinobacteriota and Firmicutes were positively correlated with pH and negatively correlated with N, and P. There was little impact on Bacteroidetes with pH, TP, AP, NO$_3^-$, and TN. At the genus level (Fig. 5B), the pH had insignificant effects on the endophytic bacteria, whereas TP, TK, TN, N, and SOC showed significant effects. TK, TP, N, SOC, and TN contributed 61.9%, 16.6%, 10.8%, 7.7%, and 3.1%, respectively. The genera with ascending abundance in the L group (compared with the DL group) were positively correlated with TP, TK, TN, N, and SOC and included *Stenotrophomonas* and *Sphingomonas*. *Chryseobacterium*, *Allorhizobium-Neorhizobium-Pararhizobium-Rhizobium* and others, which were relatively higher in abundance in the DL group than the L group, were negatively correlated with TP, TK, TN, N, and SOC.

# DISCUSSION

**Endophytic nitrogen-fixing bacteria increased in barren environments**

Nitrogen is an essential nutrient for plant growth and development and is the main component of many plant components, including chlorophyll, amino acid, and ATP (*Banik et al., 2019*). There were significant differences in soil nutrient composition between the Zoige sandy soil and the open field stock nursery in the laboratory, as shown in Table 3. The total nitrogen (16-fold difference) and available nitrogen (22-fold difference) showed significant differences between these soil types. We hypothesized that these gaps led to increased nitrogen-fixing bacteria in the DL group. *Allorhizobium-Neorhizobium-Pararhizobium-Rhizobium*, which are in the Rhizobiaceae, are among the most important nitrogen-fixing bacteria (*You et al., 2021*). A previous study showed that *Allorhizobium-Neorhizobium-Pararhizobium-Rhizobium* promoted the growth of the aboveground parts of the plant (*Pang et al., 2021*). However, they are not a well-studied endophytic bacteria. The abundance of *Allorhizobium-Neorhizobium-Pararhizobium-Rhizobium* increased in barren environments, which accounted for 6.99% in the DLS and only 1.46% in the

LS (Table S2). In addition, an association has been shown between Actinobacteriota and nitrogen-fixing bacteria. Many endophytic nitrogen-fixing bacteria were isolated in previous studies. In the phylum Actinobacteria, *Microbacterium*, *Rhodococcus*, *Nocardioides*, and *Clavibacter* increased in the DL group. *Rhodococcus* and *Nocardioides* were reported to fix nitrogen (*Kukla, Plociniczak & Piotrowska-Seget, 2014*; *Lim et al., 2014*). *Microbacterium* and *Clavibacter* did not show the ability to fix nitrogen in previous studies, but have other growth-promoting abilities, such as the ability to dissolve phosphorus (*Borah et al., 2021*; *Kumar et al., 2016*). Of the genera that increased in the DL group, *Brevundimonas*, *Chryseobacterium*, and *Paenibacillus* have also been reported to fix nitrogen (*Do Carmo Dias et al., 2021*; *Herrera-Quiterio et al., 2020*; *Naqqash et al., 2020*).

*Allorhizobium-Neorhizobium-Pararhizobium-Rhizobium* was found in all the samples we collected but was not found in rhizosphere soil (*Jian et al., 2022*). These bacteria may form symbiotic associations with *L. secalinus*, which help *L. secalinus* to adapt to low-N conditions. Other studies have not shown symbiotic associations between *Allorhizobium-Neorhizobium-Pararhizobium-Rhizobium* and *L. secalinus*, so this hypothesis remains to be directly proven. *L. secalinus* will recruit the N-fixing community as endophytic bacteria in a low N environment. The changes in endophytic nitrogen-fixing bacteria are concentrated in the roots and stems. In this study, we hypothesize that the roots and stems of *L. secalinus* are a major site of nitrogen fixation, similar to what occurs in rice (*Ito, Cabrera & Watanabe, 1980*).

## Endophytic bacteria have multiple pro-growth functions

Phosphorus is the second most significant limiting element of plant growth, which may become the first limiting element under specific ecological contexts (*Hinsinger, 2001*). Bacteria may release the precipitated P from the soil through natural solubilization (*Bhattacharyya & Jha, 2012*) and endophytic fungi can increase the host response to phosphate starvation to protect the host in the low P environment (*Hacquard et al., 2016*). TP and AP were lower in the Zoige alpine sandy soil compared to the open field stock nursery (Table 3), hindering the reconstruction of vegetation. In addition to *Microbacterium* and *Clavibacter*, *Agreia* (*Sheng et al., 2011*), *Chryseobacterium* (*Kaur et al., 2022*), and *Brevundimonas* (*Jiang et al., 2022*), whose abundance increased in the DL group, may also be able to solubilize phosphate. These genera negatively correlate with TP in RDA analysis. These phosphate-solubilizing endophytic bacteria can help *L. secalinus* to survive in the low P environment.

The other pro-growth functions also are reported in the species whose abundance increased in the DL group. *Clavibacter* (*Kumar et al., 2016*), *Rhodococcus* (*Kukla, Plociniczak & Piotrowska-Seget, 2014*), *Brevundimonas* (*Jiang et al., 2022*), and *Chryseobacterium* (*Herrera-Quiterio et al., 2020*) have IAA producing ability. IAA enhances plant growth and can increase the rate of nitrogen fixation of endophytic bacteria (*Defez, Andreozzi & Bianco, 2017*). ACC deaminase (ACCD) positive bacterial strains are present in *Rhodococcus* (*Kukla, Plociniczak & Piotrowska-Seget, 2014*) and *Polaromonas* (*Jasim et al., 2015*), which can significantly reduce reactive oxygen species (ROS) and lipid peroxidation levels (*Srinivasan et al., 2021*). In addition, ACCD can substantially increase the host's biomass,

chlorophyll content, and photosynthetic activity in low-N conditions (*Srinivasan et al., 2021*).

### Changes in the endophytic bacteria are related to the environment

There were significant differences between the endophytic bacterial community structure of *L. secalinus* under different ecological environments. Different tissues of plants in the same ecological environment also indicated some differences. An analysis of the *α*-diversity showed that even the diversity of endophytic bacteria decreased in cold, barren environments. *β*-diversity analysis showed that the endophytic bacteria in the roots were more enriched and diverse than those in the stems and leaves, suggesting that there is a closer phylogenetic proximity of stems to leaves. The soil physicochemical properties only explained <10% of the variation in the endophytic bacterial community structure, which is indirect proof of the selectivity of plants. Climate may be the primary reason for the change in endophytic bacterial communities, rather than the soil physicochemical properties.

We did not collect data on the UV index and temperature, but they are important factors affecting the endophytic bacteria. The species in which abundance increased in the DL group are tolerant of cold. For example, *Agreia* and *Polaromonas* are common genera in cold surroundings (*Franzetti et al., 2013*; *Sheng et al., 2011*), and *Polaromonas* has been shown to have UV resistance (*Ciok et al., 2018*). Some species of *Clavibacter* can significantly attenuate chilling-induced electrolyte leakage, lipid peroxidation, and the accumulation of ROS as endophytic bacteria (*Ding et al., 2011*) to improve the ability of plants to resist cold. An increased abundance of these species is strongly associated with local climatic conditions.

Functional prediction showed that the abundance of nutrient metabolism related to stress resistance increased in the DL group. This evidence illustrated that the species whose abundance increased in the DL group have a stronger ability to adapt to the environment or to promote plant growth. These species had the ability to improve the stress tolerance of *L. secalinus* to cold, strong UV, and barren soil to some extent. They have substantial potential in environmental remediation and agricultural production. However, the endophytic bacteria in *L. secalinus* in the Zoige alpine sandy soil still need to be isolated, identified, and investigated.

## CONCLUSIONS

Our study explored the changes to the endophytic bacterial community structure in *L. secalinus* under different ecological environments using 16s rRNA high-throughput sequencing. The results showed that endophytic bacteria affect different tissues of *L. secalinus* in different ecological environments. The relative abundance of the endophytic bacteria such as *Allorhizobium-Neorhizobium-Pararhizobium-Rhizobium* and their predicted functions increased in *L. secalinus* in the Zoige alpine sandy soil, indicating that they are designed to accommodate the high-altitude, cold, strong UV, and barren environment. The physicochemical properties of the soil did not explain the changes in endophytic bacteria. However, other factors, such as temperature, UV light intensity, and the selectivity of *L. secalinus*, may have a stronger influence on the endophytic

bacteria. Our study demonstrated that *L. secalinus* is a passive and active part of a differentiated endophytic bacterial community. *L. secalinus* has a limited capacity to adapt to its environment. Endophytic bacteria of *L. secalinus* in the Zoige alpine sandy soil may promote plant growth, which helps *L. secalinus* survive under harsh environments. The endophytic bacteria in *L. secalinus* in the Zoige alpine sandy soil have the potential to improve environmental remediation and agricultural production.

### Funding

This work was supported by the Second Tibetan Plateau Scientific Expedition and Research Program (No. 2019QZKK0404). The funders had no role in study design, data collection and analysis, decision to publish, or preparation of the manuscript.

### Grant Disclosures

The following grant information was disclosed by the authors:
Second Tibetan Plateau Scientific Expedition and Research Program: 2019QZKK0404.

### Competing Interests

The authors declare there are no competing interests.

### Author Contributions

- Yue Xia conceived and designed the experiments, performed the experiments, analyzed the data, authored or reviewed drafts of the article, and approved the final draft.
- Ruipeng He performed the experiments, authored or reviewed drafts of the article, and approved the final draft.
- Wanru Xu performed the experiments, prepared figures and/or tables, and approved the final draft.
- Jie Zhang conceived and designed the experiments, authored or reviewed drafts of the article, and approved the final draft.

### Data Availability

The raw sequencing reads are available in the NCBI database: PRJNA898757.

### Supplemental Information

Supplemental information for this article can be found online at http://dx.doi.org/10.7717/peerj.15363#supplemental-information.

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
