# Peer review of "The Zoige pioneer plant Leymus secalinus has different endophytic bacterial community structures to adapt to environmental conditions"

_PeerJ, doi:10.7717/peerj.15363_

## Round 0.1 · original submission · Major Revisions

The manuscript needs major revision. Please revise according to the reviewers' comments.

·

Basic reporting

The authors investigated the changes in endophytic bacterial community structure in Leymus secalinus from different ecological environments. They found the changes were significantly different, and the endophytic bacteria in L. secalinus grown in alpine sandy land may have anti-stress properties and the ability to fix nitrogen.

Experimental design

The research question was well defined.

Validity of the findings

The paper was well-organized and contained interesting information.

Additional comments

1. Authors only analyzed the endophytic bacterial community; therefore, I suggest authors should change the ‘endophytes’ to ‘endophytic bacteria’ to make it more specific. Please check it through the manuscript.
2. How the difference explains the extremely strong vitality of L. secalinus?
3. Ecological environments and tissues both affected the endophytic bacterial community structure, the differences varied? Which one conduct the greater influence?
4. The primers were different between roots and stem, leaf, why? Authors should explain it and give the reference to elaborate the design is reasonable. Otherwise, the results are unreliable.
5. Authors should explain why the diversity decreased in DL samples, but their relative abundance increased.
6. The language should be checked very carefully by an English editor since there were a lot of grammar mistakes.

Reviewer 2 ·

Basic reporting

no comment

Experimental design

no comment

Validity of the findings

no comment

Additional comments

The authors present a novel study on effects of different environments on endophytic bacterial community structure in different plant tissues of Leymus secalinus, one of the pioneer plants of Zoige desertified alpine grassland. They set 2 environmental conditions, Zoige alpine sandy land and the open field nursery in the laboratory. They used DNA extraction, PCR amplification, high-throughput sequencing and other data analyses to compare endophytic bacterial community structure. They pointed out the possible reason of anti-stress properties of Leymus secalinus in Zoige desertified alpine grassland. This study is very innovative and valuable for the future research of endophytic bacterial community of plants in alpine regions. I reviewed to the authors’ manuscripts sincerely and have a number of suggestions for improvements:

1. L22: The methods of the Abstract should state clearly which ecological environments were set up for the experiment.
2. L46-48: “Owing to climate change and anthropogenic factors, degradation of the grasslands …”, What are the specific climate change and anthropogenic factors?
3. L48: “Fortunately, desertification was effectively controlled …”, The sudden mention of desertification is not very precise. Is the "degradation of the grasslands" in the previous sentence desertification?
4. L49: “the establishment of biological sand barriers and the current prohibition on grazing” Can these factors bring desertification under control? Is prohibition on grazing the reason desertification is under control? Compared to other forms of human disturbance or abiotic factors such as climate change, who contributes more to control of grassland desertification? If you can prove the statement of this sentence, or the exact effect of grazing on desertification, please quote it at the end of the sentence.
5. L65: The endophytes, from the stress resistance of Leymus secalinus in the last paragraph, are suddenly mentioned in this paragraph, which is very abrupt. Authors should add a sentence or two about the transition from stress resistance to endophytes. For example, explaining what role endophytes might play in stress resistance.
6. L74-79: This sentence should be interchangeable with the previous one. There is no logic in this paragraph, and the order of sentences needs to be adjusted to make paragraphs and sentences coherent, so as to better introduce the lack of current research in this field and increase the readability of the paper. For example, by foreshadowing and eventually leading to “but the plants used in phytoremediation in the Qinghai Tibet Plateau have not been adequately studied.”
7. L96: “and the samples collected in the laboratory”, it should be rewritten as: “and the samples collected in the open field nursery in the laboratory”. In addition, the treatment should appear in the Abstract.
8. The most important future improvement in this paper is to enhance the logic of the description. In addition, the result is reasonable, and the conclusion is valuable, the figure is very good.

Annotated reviews are not available for download in order to protect the identity of reviewers who chose to remain anonymous.

---

## Round 0.2 · Minor Revisions

Linguistic improvement is needed throughout the manuscript.

·

Basic reporting

The authors answered correctly. The paper was much better.

Experimental design

Well defined.

Validity of the findings

Conclusions are well stated.

Additional comments

Language should be checked carefully, especially the revised contents.

Reviewer 2 ·

Basic reporting

The manuscript has the professional article structure, figures, tables.

Experimental design

The experiment design of this manuscript belongs to original primary research.

Validity of the findings

Conclusions are well stated, linked to original research question & limited to supporting results.

Additional comments

The authors have responded to the comments in detail and made changes to the manuscript. We have reviewed the manuscript again in detail. We suggest "accept". Thank you very much!

---

## Round 0.3 · accepted · Accept

Based on the revised submission, the reviewers endorsed the manuscript for publication.

·

Basic reporting

The paper is much better.

Experimental design

It's fine.

Validity of the findings

Fine.

Additional comments

No additional comments.